# Damage Detection of Unwashed Eggs through Video and Deep Learning

**DOI:** 10.3390/foods12112179

**Published:** 2023-05-29

**Authors:** Yuan Huang, Yangfan Luo, Yangyang Cao, Xu Lin, Hongfei Wei, Mengcheng Wu, Xiaonan Yang, Zuoxi Zhao

**Affiliations:** 1College of Engineering, South China Agricultural University, Guangzhou 510642, China; huang_yuan@stu.scau.edu.cn (Y.H.); fit@stu.scau.edu.cn (Y.L.); caoyangy@stu.scau.edu.cn (Y.C.); lx_zxc2156007@stu.scau.edu.cn (X.L.); hongfei.wei@keystarirobot.com (H.W.); 13119046520wmc@stu.scau.edu.cn (M.W.); 20223175020@stu.scau.edu.cn (X.Y.); 2Key Laboratory of Key Technology on Agricultural Machine and Equipment, South China Agricultural University, Ministry of Education, Guangzhou 510642, China

**Keywords:** crack detection, YOLOv5, egg processing, ByteTrack, unwashed egg

## Abstract

Broken eggs can be harmful to human health but are also unfavorable for transportation and production. This study proposes a video-based detection model for the real-time detection of broken eggs regarding unwashed eggs in dynamic scenes. A system capable of the continuous rotation and translation of eggs was designed to display the entire surface of an egg. We added CA into the backbone network, fusing BiFPN and GSConv with the neck to improve YOLOv5. The improved YOLOV5 model uses intact and broken eggs for training. In order to accurately judge the category of eggs in the process of movement, ByteTrack was used to track the eggs and assign an ID to each egg. The detection results of the different frames of YOLOv5 in the video were associated by ID, and we used the method of five consecutive frames to determine the egg category. The experimental results show that, when compared to the original YOLOv5, the improved YOLOv5 model improves the precision of detecting broken eggs by 2.2%, recall by 4.4%, and mAP:0.5 by 4.1%. The experimental field results showed an accuracy of 96.4% when the improved YOLOv5 (combined with ByteTrack) was used for the video detection of broken eggs. The video-based model can detect eggs that are always in motion, which is more suitable for actual detection than a single image-based detection model. In addition, this study provides a reference for the research of video-based non-destructive testing.

## 1. Introduction

Eggs are a staple food in people’s daily life. They are rich in protein, lipids, and various vitamins and are indispensable in the human diet [1]. Due to the brittle and sensitive nature of eggshells, they are easily damaged in various ways during production and transportation. According to statistics, 3% of all eggs are broken (including breakage, cracking, or holes in the eggshell) [2]. Broken eggs are not only at an increased risk of being infected by bacteria, such as *Salmonella*, making eggs harmful to humans, but also easily contaminate other intact eggs and processing production equipment during transportation [3,4].

In several countries, including Australia, the US, and some European nations, eggs are subjected to washing before being offered for sale. However, in all Asian countries other than Japan, egg production involves unwashed eggs [5,6]. During the sterilization and cleaning of eggs, the liquid from damaged eggs can easily leak out and contaminate the disinfection pool [7]. Furthermore, washed eggs have a shorter shelf life, which is the primary reason why they may become rotten before being sold [8,9]. For the automatic detection system, the presence of dirt on the surface of unwashed eggs presents a greater challenge to detection and easily leads to false detection by the system.

In order to meet people’s increasing demand for egg quantity and quality, researchers and technicians continue to carry out research into the automatic detection of broken eggs. The system they developed is primarily designed to detect breakages on washed eggs. Among them, the detection system can be mainly divided into methods based on acoustic analysis and methods based on machine vision. The method based on acoustic analysis is mainly used to determine whether the egg is damaged by analyzing the frequency response of mechanical vibration applied to the egg. However, the uneven thickness of the eggshell and the surrounding environment can easily affect the accuracy of acoustic detection [2,5,10]. The vision-based method mainly analyzes the acquired image through image processing technology or machine vision technology to judge whether the egg is broken. In recent years, machine vision-based methods have made breakthroughs in broken egg detection.

A neural network combined with wavelet transformation for egg detection and classification was proposed by Peng [11]. The accuracy of detecting intact and cracked eggs reached 95% and 90%, respectively. Wang et al. [12] reconstructed and decomposed the egg image through dual-scale wavelet transformation and obtained a 24-dimensional parameter system. Then, principal component analysis (PCA) was used to reduce the dimensionality of the parameter system to obtain three characteristic parameters that can be used for the detection of broken eggs, and the accuracy of this method reached 96%. Priyadumkol [5] designed a machine vision-based detection system. The system can analyze the images collected in the vacuum pressure chamber and detect broken eggs by combining a Gaussian feature and a median feature. The results showed a broken egg detection accuracy of 94%, and the false positive rate was 1.67%. Bao et al. [13] proposed a method to use a Laplacian-Gaussian operator (LOG) to enhance the crack on an egg and ensure the continuity of this crack by using an improved LFI (local fitting image) index to distinguish the crack area. According to their experimental results, the accuracy reached 92.5%. Dehrouyeh et al. [14] proposed an algorithm based on the HIS color space for the detection of broken eggs and eggshell fouling. Among them, the maximum value at both ends of the histogram was used as the detector of broken eggs, and the connected domain detection technology was used to detect eggshell dirt. Their experimental results showed that the detection accuracy of broken eggs and intact eggs was 90.66% and 91.33%, respectively. Xiong et al. [15] used SVM to classify broken eggs and intact eggs and proposed a model with 13 feature parameters mixed within five different domains. Their experimental results show that the accuracy of the model reached 97.5%. Omid et al. [16] developed an intelligent egg-grading system by combining fuzzy logic and machine vision technology. The experimental results showed that this method obtained 94.5% accuracy in broken egg detection

Computer vision technology has been widely used in agriculture due to its advantages of accuracy and speed. Among them, the deep learning algorithm based on image classification has been more and more researched in the detection of broken eggs. Datta et al. [17] proposed a method for detecting eggs with severe streaks and cracks using Faster R-CNN. Their experimental results show that the mean accuracy precision (mAP) of the model is greater than 75%, and this method can detect most of the damaged areas. Later, Tang et al. [18] used an improved MobileNetV3_large to detect broken preserved eggs with an accuracy rate of 96.3%. Turkoglu [19] combined CNN and BiLSTM to detect defective eggs. Their experimental results showed that the accuracy of this method is close to 100% for the detection of intact eggs and dirty eggs, and 96.3% for the detection of broken eggs. Omid et al. [20] proposed a method based on an improved VGG16 for egg classification. Through five-fold cross-validation, the average overall accuracy of the model was 94.84%. Botta et al. proposed an automatic eggshell crack detection method based on a convolutional neural network (CNN) [21], using an SVM classifier trained using a histogram of gradient orientation (HOG) and a local binary pattern (LBP). The segmented egg image blocks were classified, and their experimental results show that the accuracy rate reached 95.38%.

There are several problems in the studies of vision-based broken egg detection found in the literature. First of all, there is no rotating device designed to show the entire surface of the egg, and the damaged area of the egg can be located at the bottom, which makes it easy to falsely detect. Secondly, when the image classification algorithm is used to classify eggs, it is only suitable for applications in static scenes and is not suitable for the dynamic scenes of actual production lines. Finally, the proposed method is not suitable for large-scale detection. This study aims to propose broken detection on unwashed eggs by using video scenarios, which are used to achieve the large-scale detection of eggs in dynamic situations. In this study, a visual inspection system is firstly designed on the basis of the six-channel egg-conveying device, and images are collected on the actual production line. Then, we put forward a suitable detection method according to the actual movement of the egg. Finally, the method was verified by video.

The remainder of this study is organized as follows. Section 2 introduces the detection system and the proposed method, as well as the dataset used to train the model. Section 3 introduces the experimental work. It includes a comparison of the performance of different models in the detection of broken eggs and the detection performance of the models in practical applications. Finally, Section 4 and Section 5 discuss the findings and conclude, respectively.

## 2. Materials and Methods

### 2.1. Machine Vision System

In this study, the device shown in Figure 1 was used to transport eggs, which consists of a conveyor belt and a control box. Among them, the control box can control the transmission speed of the conveyor belt and the opening and closing of lights. The conveyor belt is mainly used for the transmission of eggs and contains a total of six transmission channels. There are many rollers on each channel, which form compound movements regarding rotation and translation under the action of the motor. The force between the eggs and the roller makes the eggs undergo a compound movement of rotation and translation on the conveyor belt. Each channel of the conveyor belt can convey 3600 eggs per hour; that is, the device can convey 21,600 eggs per hour.

On the basis of the conveying device, a machine vision system (as shown in Figure 2) is designed. Considering the color of the eggs and the possibility of reflection, the system uses a ring LED light as the light source. When the camera is working, the imaging plane of the camera is parallel to the horizontal movement plane of the egg. In order to display as many eggs as possible in the image, the height direction of the image is parallel to the horizontal movement direction of the eggs in the field of view. As the egg rotates past the camera’s field of view, the entire surface of the egg is captured. The image acquisition work of this study and the follow-up verification work are all carried out on this system.

### 2.2. Production of Datasets

In order to make a dataset for training YOLOv5 and make the dataset more suitable for the actual working conditions, we took photos of unwashed eggs on an egg processing production line of the Gonghe Group in Guangzhou, Guangdong Province. In this study, the A3200CU000 industrial camera produced by Dahua was used for image acquisition, which has a maximum resolution of 1920 × 1080 and a frame rate of 120 fps (frames per second). 

In the actual production line, because the proportion of broken eggs in the total number of eggs is not high, two ways to obtain images were adopted for this paper in order to ensure the balance of image samples. A camera was directly used to capture images of the eggs taken out from the conveyor belt in method 1; each egg was rotated 120° around the long axis to take single pictures. A total of three photos were taken of one egg, each of which contained only one egg, and the photos were mainly of broken eggs. In method 2, we used a camera to shoot video of the eggs on the conveyor belt and set the speed of conveying eggs to 21,600 eggs per hour; the shooting frame rate was set to 120 fps, and the resolution was set to 1920 × 1080. During the transmission process, the maximum number of eggs in the field of view is 42. Images were extracted every 120 frames from the collected video, and the images collected by this method contained multiple eggs. Finally, a total of 3726 pictures of unwashed eggs were obtained through methods 1 and 2. Among them, a total of 3448 pictures were collected through method 1, and a total of 278 pictures were collected through method 2. The number of intact eggs in all pictures reached 7936, and the number of broken eggs reached 6532.

After that, we divided the eggs into two categories: non-broken eggs (intact eggs) and broken eggs (breakage, cracking, or holes in the eggshell). These two categories of eggs have a certain probability of being combined with dirt (including organic residues, such as droppings, blood, and yolk). The egg samples are shown in Figure 3.

After image acquisition, we used the annotation tool LabelImg to annotate the image dataset; the image annotation process is shown in Figure 4. Finally, the acquired images and annotation information were combined into the dataset, and the dataset was divided into a training set, a verification set, and a test set according to the ratio of 7:2:1 for training YOLOv5. 

### 2.3. Broken Egg Detection Methods

The method in this study can detect broken eggs in dynamic scenes. The process of this method is as follows: (1) adopt the improved YOLOv5 to detect the video frames acquired by the camera to obtain the current categories and positions of all eggs. (2) Adopt the ByteTrack tracking algorithm to track eggs and assign an ID to each egg. (3) Use the discrimination method of continuous frames to judge the category of eggs in motion, and complete the counting of eggs.

#### 2.3.1. Broken Egg Detection Based on Improved YOLOv5

YOLO (you only look once) is a one-stage object detection algorithm based on regression [22]. YOLOv5 is the fifth version of the YOLO series. When compared with the two-stage algorithm, it removes the region proposal network and greatly improves the detection speed. YOLOv5 can be divided into three parts: backbone, neck, and head. New-cspdarknet53 integrates with the SPPF module as the backbone, while New-cspdarknet53 is mainly used for feature extraction, and the SPPF is an improvement on the SPP (spatial pyramid pooling). To a certain extent, SPPF can improve the efficiency of feature extraction and solve the problem of multi-objective scale. In the neck part, PAN (path aggregation network) is adopted, which can enhance the neck’s feature fusion ability, thereby improving the detection ability of different objects and improving the propagation of the underlying features. The head part consists of three Detect detectors that perform detection on feature maps of different scales. YOLOv5 includes four network models with different performance: YOLOv5s, YOLOv5m, YOLOv5l, and YOLOv5X. In these four versions, the width and depth and the detection accuracy of the networks increase gradually while the detection speed decreases gradually [23]. In this study, in order to balance the accuracy and speed of detection, we use the improved YOLOv5m to process the images collected by the camera to obtain the category and detection box of each egg in each frame.

When detecting broken eggs on the conveyor device, on the one hand, the translation and rotation of the eggs will cause motion blur; on the other hand, due to the large number of eggs in each image, each egg does not occupy a high number of pixels, so the information used to identify damage only occupies a small pixel area. In the process of downsampling the image, the information is not easily extracted by the network, which may lead to the misidentification of eggs. In order to improve the detection accuracy, YOLOv5 is improved in the following ways.

1.Co-ordinate Attention (CA)

The attention mechanism originated from the RNN model was applied to NLP and has also been applied in the image field in recent years. The attention mechanism can suppress the expression ability of non-essential features on the feature map and enhance the expression ability of the main features [24]. CA (Co-ordinate Attention) is an attention mechanism that can embed position information into channel attention [25]. It can not only avoid the loss of position information due to global pooling but can also enhance the expressive ability of mobile network information features. The CA module can transform any intermediate tensor $X=\left[x_{1},x_{2},\ldots ,x_{c}\right]$ in the network and output a tensor $Y=\left[y_{1},y_{2},\ldots ,y_{c}\right]\in R^{H\times W\times C}$ of the same size. Figure 5 presents the CA module network structure.

Firstly, the CA module divides the input feature map into two directions: width and height, and global average pooling is performed to obtain the feature map, respectively. The average pooled feature map is then subjected to the concat operation and 1 × 1 convolution. After that, the batch normalized feature map is sent to the Sigmoid activation function to obtain a 1 × (*H* + *W*) × *C*/*r* size feature map. The feature map is convoluted by 1 × 1 according to the height and width directions to obtain the feature map with the same number of channels as the original, and then the attention weights in the height and width directions of the feature map are obtained through the sigmoid activation function. Finally, the weights are multiplied on the original feature map to obtain the feature map with attention weights in the width and height directions. 

CA can avoid the loss of position information caused by global pooling and effectively use the spatial co-ordinate information of the input feature map. In the shallow network of YOLOv5, due to the large spatial feature map and the small number of channels, the weights of the channels obtained cannot summarize the specific features. In the deep network, too many channels can easily lead to overfitting. Therefore, this study added CA to the last layer of the backbone of YOLOv5 to improve the model’s ability to locate and identify objects.

2.BiFPN

In early single-stage detection algorithms, single-stage feature maps cannot effectively represent objects of various scales simultaneously. Therefore, the feature pyramid network (FPN) was developed to represent objects at different scales [26]. The FPN introduces a top-down channel to fuse features. It takes the features of different resolutions generated in the previous step as input and outputs the fused features. It can also solve the multi-scale problem in object detection and greatly improve the detection performance of small objects without increasing the amount of calculation of the original model. Based on FPN, the path aggregation network (PANet) is proposed [27]. PANet adds a bottom-up pyramid behind FPN to realize the two-way fusion of top-down and bottom-up so as to enhance the transmission of low-level positioning features. 

The proposal of PANet proves the effectiveness of two-way fusion. BiFPN is a two-way fusion network that uses a fast normalization algorithm to assign different weights to the features of each layer and adjusts the input contribution to the output feature maps through the weights [28]. On the basis of PANet, BiFPN adds an additional channel between the input and output nodes to fuse more features and deletes the intermediate node with only one input edge to simplify the network structure. Figure 6 shows the structure of FPN, PANet, and BiFPN. 

In order to enhance the feature fusion ability of the YOLOv5 neck, it is adapted to the network fusion of different input features, improving the multi-scale recognition ability and small-object-recognition ability of YOLOv5; BiFPN is used in this study to replace the PANet structure in YOLOv5.

3.GSConv

In general, a model with more neurons can obtain stronger non-linear expressiveness. However, an increase in model complexity is often accompanied by a decrease in speed. The lightweight design of the network implemented by the depthwise separable convolution (DSC) operation can effectively reduce the model parameters and FLOPs, but DSC is weaker than standard convolution (SC) in feature extraction and fusion capabilities.

In order to make the model more suitable for large-scale rapid detection, it is necessary to reduce the parameters of the improved YOLOv5 and reduce the computational cost; this study incorporated the lightweight network GSConv into YOLOv5 [29]. The structure diagram of GSConv is shown in Figure 7. GSConv combines DSC and SC and uses shuffle to infiltrate the information generated by SC into every part of the information generated by DSC, which can overcome the separation of DSC in the process of processing input image channel information and improve the feature extraction and fusion ability.

In this paper, GSConv was added to the neck part of YOLOv5 to minimize the loss of semantic information caused by the spatial compression of the feature maps and channel expansion. The improved YOLOv5 structure is shown in Figure 8. 

#### 2.3.2. Egg Tracking Based on ByteTrack 

In order to prevent false detection caused by the damaged area of the egg located on the back of the egg, it is necessary to apply rotation to the egg during the detection process. For a broken egg, due to the rotation of the egg, the damaged area is in a cycle of being visible and invisible. At this time, the same egg has different detection results in different frames. In other words, only using target detection and image classification algorithms cannot reflect the actual category of eggs in dynamic scenes. In order to accurately determine the category of eggs during egg movement, this study uses ByteTrack [30] to track the eggs within the camera’s field of view, which is used to correlate the detection results of YOLOv5 at different times.

ByteTrack is a detection-based tracking (DBT) algorithm. When using ByteTrack to track eggs, it completes egg tracking by creating Tracks that correspond to eggs, matching them with YOLOv5’s detection results frame by frame. Among them, Tracks represents the information of the track, and the workflow of ByteTrack is shown in Figure 9. ByteTrack divides the YOLOv5 detection boxes into high-scoring boxes and low-scoring boxes according to the confidence score, and divides Tracks into long-term Tracks and short-term Tracks. The long-term Tracks will be matched with the high-scoring boxes after using the Kalman filter [31] for prediction. The long-term Tracks that are successfully matched will directly use the corresponding high-scoring boxes for position update, and the long-term Tracks that fail to match will be matched with the low-scoring boxes. The unmatched high-scoring boxes are matched with the short-term Tracks after Kalman filter prediction.

Before the ByteTrack algorithm was proposed, the DBT algorithm only used the high-scoring detection box, while the low-scoring detection box was directly discarded, resulting in the loss of the real target and interruption of the trajectory. ByteTrack proposed a simple and effective method to improve the performance of the tracker by associating each detection box and made the low-score detection box participate in the tracking, instead of simply eliminating the low-score detection box. When compared to other DBT algorithms, ByteTrack effectively reduces the problem of transient target disappearance caused by target occlusion, motion blur, and other problems in the tracking process, and the problem of trajectory breaking and ID switch caused by low target confidence. In addition, it also has excellent performance in tracking speed.

#### 2.3.3. Real-Time Video-Based Detection Model

In this study, the improved YOLOv5 was used to detect broken eggs in images captured by the camera, and the detection box and category of each egg in the current frame were obtained. In addition, ByteTrack was used to track the eggs currently within the camera’s field of vision so as to correlate the detection results of the same egg at different times. 

As shown in Figure 10, the red boxes represent broken eggs, and the green boxes represent intact eggs. When the egg moved within the camera’s field of vision, the damaged area rotated from the bottom of the egg to the top and then to the bottom. At this time, the detection result of YOLOv5 changed from a non-broken egg (intact egg) to a broken egg and then to a non-broken egg (intact egg). Obviously, using YOLOv5 alone at this point cannot accurately determine the actual category of the egg. The tracking algorithm can assign the same ID to the same egg in different time periods. Based on this relationship, we assign a storage space to each ID, which is used to store the results detected by YOLOv5 in different time periods; only when an egg is judged as being a broken egg by YOLOv5 for five consecutive frames can it be judged as a broken egg by our model. When an egg is judged as broken, even if the damaged area is located at the bottom of the egg, our model can still judge the actual category of the egg.

### 2.4. Evaluation Metrics

In order to verify the effectiveness of our method, we use precision, recall, FPS, and mean average precision (mAP) to evaluate the model for object detection algorithms. The calculation of these indicators [32] is shown in the following Equation (2):(1)$$Precision=\frac{TP}{TP+FP}$$
(2)$$Recall=\frac{TP}{TP+FN}$$
(3)$$AP=\int_{0}^{1}Precision\cdot Recall\;dr$$
(4)$$mAP=\frac{\sum_{i=1}^{n}AP_{i}}{N}$$
where *TP* (true positives) represents the number of positive samples predicted by the model as positive, and *FN* (false negatives) represents the number of positive samples predicted by the model as negative; *FP* (false positives) represents the number of negative samples predicted by the model as positive. *N* is the number of categories, which is equal to 2 in this study.

## 3. Results

### 3.1. Model Training Results and Ablation Experiments

The training environment adopted in this study is shown in Table 1. When training the network, the batch size is set to 16, and the image resolution of the input network training is set to 960 × 960. The SGD (stochastic gradient descent) is used to optimize the network model parameters during training. The initial learning rate, momentum, weight decay, and other parameters use the original parameters in YOLOv5, and the learning rate is adjusted using the cosine annealing process. The model runs on a GPU for 200 epochs in total.

After 200 epochs of model training, the change curve of each index and loss function is obtained, as shown in Figure 11. From the figure, it can be seen that in the first 70 epochs of training, all losses rapidly decreased, and all indicators rapidly improved. When the epochs reached 100, the model began to converge, and the curve gradually flattened. Finally, 200 epochs of training were selected to end the training.

In order to verify the effectiveness of the improved model, an ablation experiment was carried out on YOLOv5_add. In this study, the original YOLOv5 was used as the basic network, CA was added to the backbone, GSConv was added to the neck, and the BiFPN network structure was used in the neck to conduct experiments. The results of the ablation experiments are shown in Table 2, where “√” and “−” stand for the method we choose and do not choose, respectively.

It can be seen from Table 2 that various improvement methods have improved the performance of the model to varying degrees. The results show that after adding CA to the backbone, the mAP of the model can reach 93.4%, indicating that after adding CA, the model can more accurately extract the egg damage information from the image. By integrating BiFPN and GSConv into the neck, we can improve the feature expression ability and multi-scale recognition ability of the network and further improve the detection performance of the model. After adding the three improvements to the model at the same time, the mAP of the model increased by 4.1%, which verified the effectiveness of the improvement of this study.

### 3.2. Comparison of Object Detection Algorithm

In order to further verify the detection performance of improved YOLOv5 on broken eggs, this study trained SSD [33], Faster RCNN [34], YOLOv4 [35], YOLOv7 [36], and YOLOv5_add under the same dataset and compared them. Taking Precision, Recall, mAP, and FPS as the performance indicators of the model, the result comparison is shown in Table 3.

As can be seen from Figure 12, the indicators of the improved model are better than SSD, Faster RCNN, YOLOv4, and the original YOLOv5. Among them, the improved model is 2.2% higher than the original YOLOv5 model in precision, 4.4% higher in recall, and can detect broken eggs more accurately. Due to the addition of the GSConv module, the size of the improved model is reduced. When compared with the original model, the size of the improved model is reduced by 12.5%, the model is reduced by 5.1M, and the detection speed is increased by 15.4%. Although the improved model has a lower precision than YOLOv7, it has a higher recall rate and mAP:0.5 and a faster detection speed. Among them, the FPS of the improved YOLOv5 reaches 45, which is more suitable for real-time detection on the actual production line.

### 3.3. Comparison of Actual Detection Effect

In this study, the original YOLOv5 and the improved YOLOv5 were used to detect the same picture to test the effect of the improved YOLOv5 in actual detection. The results are shown in Figure 13, where the eggs circled in red are false detections of the original YOLOv5.

In Figure 13, the red boxes represent broken eggs, and the green boxes represent intact eggs. It can be seen from the figure that both the original YOLOv5 and the improved YOLOv5 can accurately detect intact eggs, but the improved model has a higher confidence. When detecting broken eggs with obviously damaged areas, both the original YOLOv5 and the improved YOLOv5 have ideal detection results, but when the damaged areas are subtle, the original YOLOv5 cannot accurately judge the category of eggs. The improved YOLOv5 has better detection performance than the original YOLOv5 due to its stronger feature extraction and feature fusion capabilities, as well as higher accuracy in identifying eggs and better performance in detecting broken eggs.

### 3.4. Application Performance

This study adopts the combination of improved YOLOv5 and ByteTrack to realize the detection of broken eggs in dynamic scenes. In order to evaluate the effectiveness of this method in practical detection, we conducted a video-based broken egg experiment on the proposed machine vision system. During the experiment, we set the camera resolution to 1920 × 1080, the frame rate to 30, the egg transmission speed to 21,600 per hour, and the egg can rotate for 2 weeks within the field of view of the camera. As the egg enters and leaves the camera’s field of view, it is only partially captured by the camera. In order to prevent false positives from our detection system, we do not detect and track incoming and outgoing eggs. In addition, we realized the automatic counting of eggs during the experiment. The detection results and effects are shown in Table 4 and Figure 14.

It can be seen from Table 4 that when using the model proposed in this study to detect intact eggs and broken eggs in dynamic scenes, the precision is 97.3% and 96.7%, the recall is 95.1% and 96.0%, and the overall detection accuracy is 96.4%. In Figure 14, the red box and the green box, respectively, represent the broken egg and the intact egg that our model currently considers. In the process of egg movement, our method can assign an ID to each egg and assign a storage space to each ID to store the detection results of the improved YOLOv5 during the egg rotation process. For broken eggs, only when a certain number of consecutive frames are detected as broken eggs by the improved YOLOv5 will it be determined as a broken egg by our model, and when the damaged area is rotated to be invisible, our model can still judge the category of the egg.

In this study, five consecutive frames are used to judge the category of eggs, and the speed of the conveyor is 21,600 eggs per hour. Figure 15 and Figure 16, respectively, show the detection accuracy of the model using different consecutive frames and different transmission speeds. As shown in Figure 15, as the number of consecutive frames increases, the accuracy of the model first increases and then decreases. When using lower continuous frames, improving YOLOv5’s misdetection of intact eggs as broken eggs is more likely to cause misdetection in the model; that is, the detection results of the model are more inclined toward broken eggs. When using larger continuous frames, only the improved YOLOv5 continuous multi-frame false detection model will have false detections; that is, the model detection results are more inclined toward the intact eggs. When using five consecutive frames to distinguish eggs, the accuracy of the model is the highest, reaching 96.4%, which is more suitable for the machine vision detection system proposed in this study. When detecting broken eggs at different transmission speeds, when the transmission speed is low, the accuracy of the model will be improved due to the reduction in the impact of egg motion blur, but when the transmission speed drops to a certain value, the accuracy will no longer increase because, at this time, it is no longer motion blur that mainly affects the accuracy of the model but the detection performance of YOLOv5.

## 4. Discussion

At present, there are mainly two methods for detecting broken eggs based on machine vision and based on acoustics. Acoustic-based detection methods can identify cracks more easily and accurately, but the accuracy may be disturbed by ambient noise, and expensive equipment is required, so the application in small and medium-sized egg production enterprises is relatively limited. The vision-based detection method has the advantages of high speed and low equipment cost. Vision-based broken egg detection methods mainly use image processing, deep learning, or machine learning, some of which are shown in Table 5, where “√” and “−” represent the types that contain and do not contain. In the vast majority of studies, the detection of broken eggs is carried out on washed eggs, and researchers often only consider the accuracy of the method but do not design the method according to the detection status in the actual egg production line.

Most studies using machine vision-based methods to detect the breakage of eggs suffer from the following two problems. The first problem is the device that can show the whole surface of the egg is not designed; in other words, the problem of false detection when the damaged area of the egg is located on the back of the egg is not solved. The second is the problem of detection speed. As shown in Table 5, most studies use classification models to discriminate egg categories. The detection time of the machine vision system is mainly composed of the time for the vision system to process the image and the time required for the system to transport the egg to the camera’s field of view for shooting. The detection process adopted by most researchers is to start the transmission device to transmit the egg to the camera’s field of view; then, the transmission device stops working, and the camera takes a picture and uses the image classification network to classify the eggs in the image. Finally, the conveyor starts up again, and the eggs are transported away. In the process of processing, the transmission device is always in the cycle of start and stop, which greatly reduces the detection speed of the system. 

When compared to current research, the method proposed in this study combines object detection and object tracking and has higher accuracy in the detection of unwashed broken eggs. In addition, this method has the advantages of low equipment cost, being able to dynamically judge the category of eggs and process multiple eggs simultaneously, making it more suitable for large-scale egg detection and promoting the application of vision-based detection methods in actual production lines. The method proposed in this study is completely applicable to other devices that can make eggs rotate and translate. However, since the transmission speed and rotation speed of other researchers or enterprise transmission equipment may be different from the device in this study, achieving the best detection effect may require fine-tuning the speed. In addition, this method still has problems, such as an insufficient ability to recognize microcracks that are difficult to observe with human eyes and failure to detect blood eggs. In future research, we will use bottom backlighting to detect blood eggs and broken eggs and further reduce the size of the model to facilitate actual deployment.

## 5. Conclusions

This study proposes a vision-based broken egg detection method for unwashed eggs. We build a machine vision system on the basis of a device that can make the egg rotate and translate. Among them, the improved YOLOv5 is used to detect video frames, and ByteTrack is used to track eggs so as to realize the detection of broken eggs in dynamic scenes. The experimental results show that the performance of the improved YOLOv5 in detecting broken eggs is better than that of the original YOLOv5. In addition, the proposed method is verified by video, and the results show that the detection accuracy of the proposed method reaches 96.4%, which meets the real-time detection requirements. The method proposed in this study provides a suitable technical solution for the large-scale rapid detection of egg quality and other food surface defects. Future research will focus on improving detection accuracy and detecting other defect categories in eggs.

## Figures and Tables

**Figure 1 foods-12-02179-f001:**
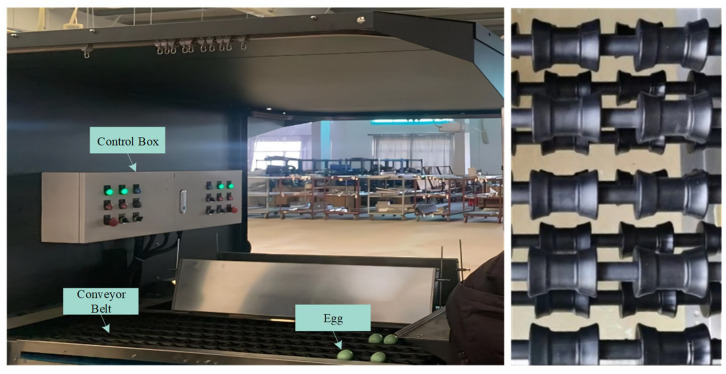
Egg conveying device.

**Figure 2 foods-12-02179-f002:**
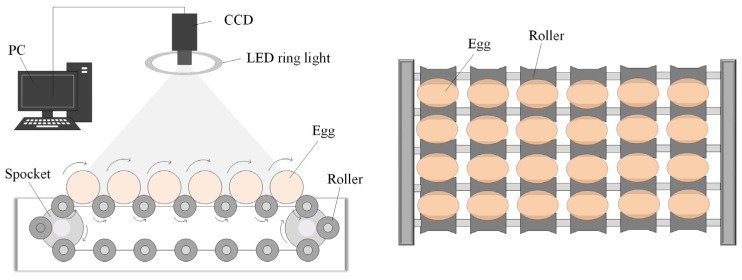
Proposed machine vision system.

**Figure 3 foods-12-02179-f003:**
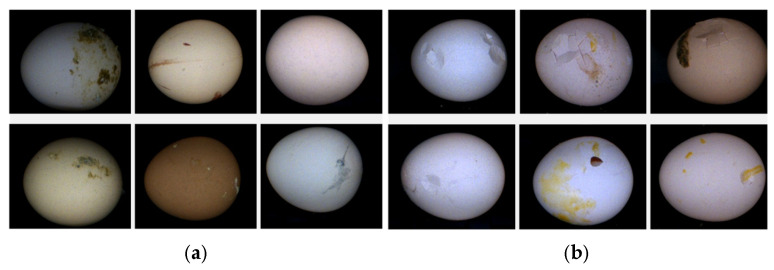
Different categories of egg samples. (**a**) Non-broken egg; (**b**) broken egg.

**Figure 4 foods-12-02179-f004:**
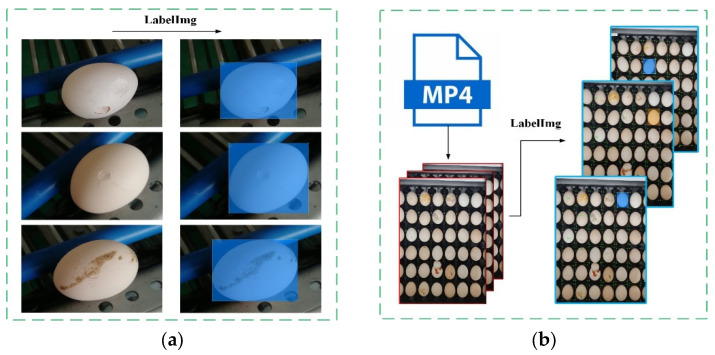
Part of the annotation data. (**a**) Annotation of the image obtained by method 1; (**b**) annotation of the image obtained by method 2.

**Figure 5 foods-12-02179-f005:**
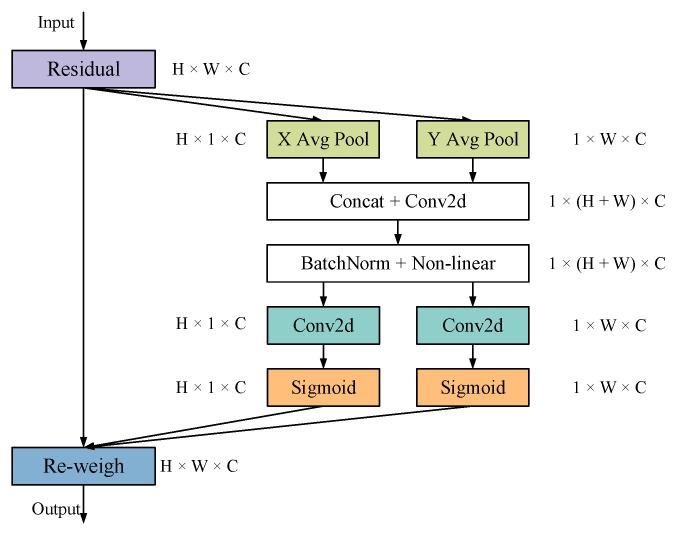
The structure of the CA module.

**Figure 6 foods-12-02179-f006:**
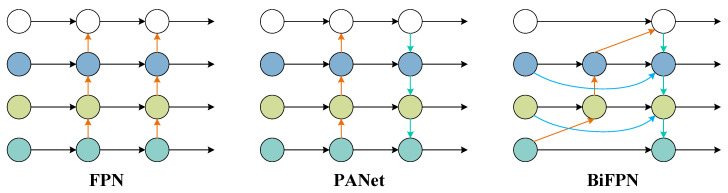
The structure of FPN, PANet, and BiFPN.

**Figure 7 foods-12-02179-f007:**
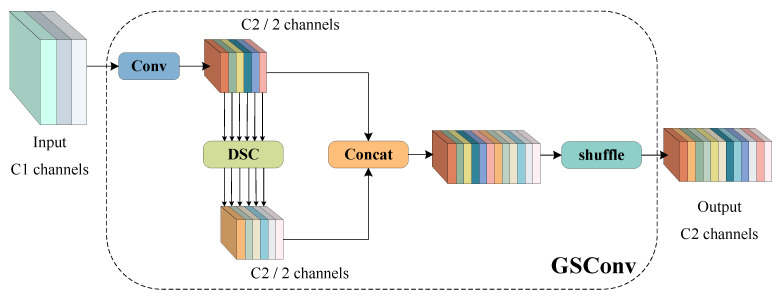
The structure of GSConv.

**Figure 8 foods-12-02179-f008:**
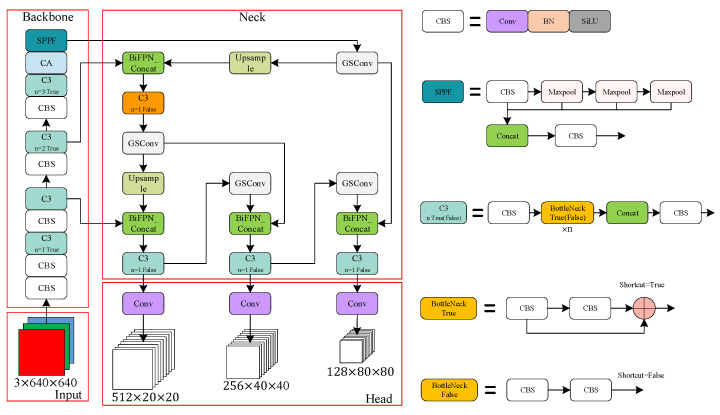
The structure of YOLOv5_add.

**Figure 9 foods-12-02179-f009:**
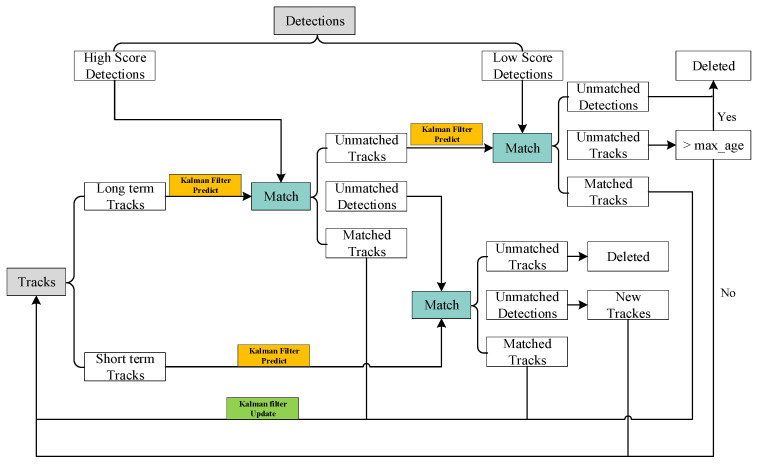
The structure of YOLOv5m_add.

**Figure 10 foods-12-02179-f010:**
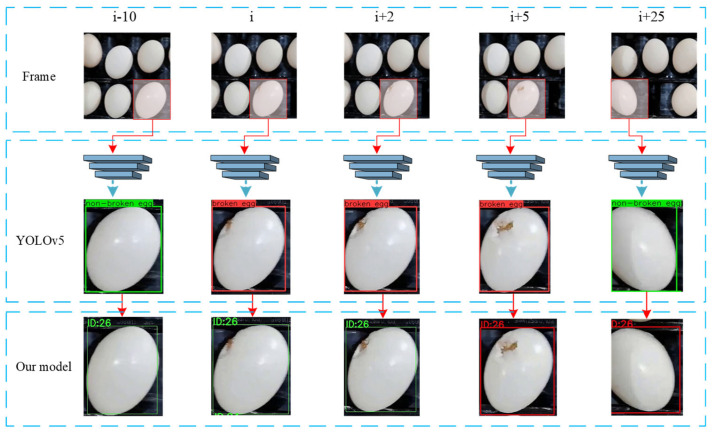
Method of judging broken eggs.

**Figure 11 foods-12-02179-f011:**
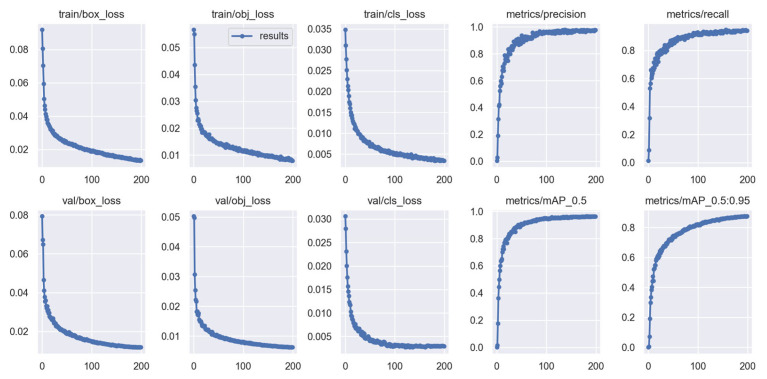
Training results of YOLOv5_add.

**Figure 12 foods-12-02179-f012:**
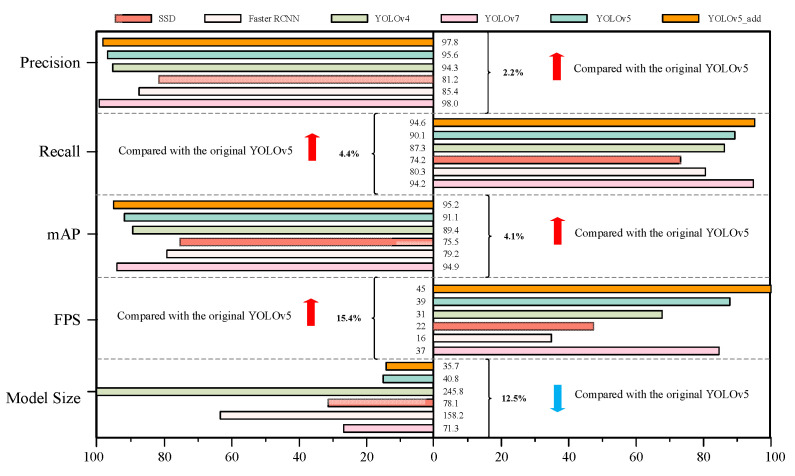
Comparison of YOLOv5_add and other models in detection performance and speed.

**Figure 13 foods-12-02179-f013:**
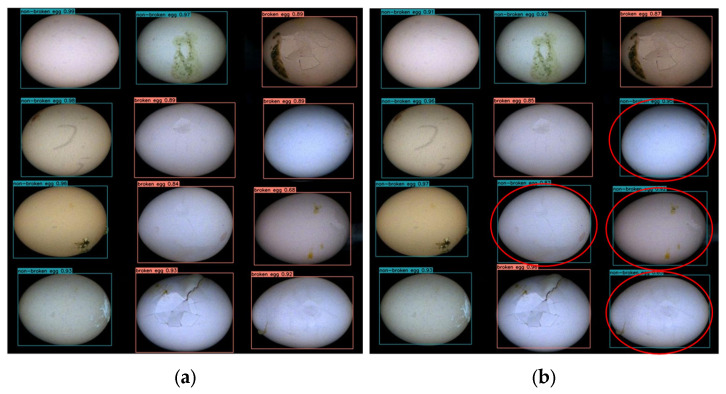
Comparison of the actual detection effect of the original YOLOv5 and the improved YOLOv5. (**a**) Improved YOLOv5 detection results. (**b**) Original YOLOv5 detection results.

**Figure 14 foods-12-02179-f014:**
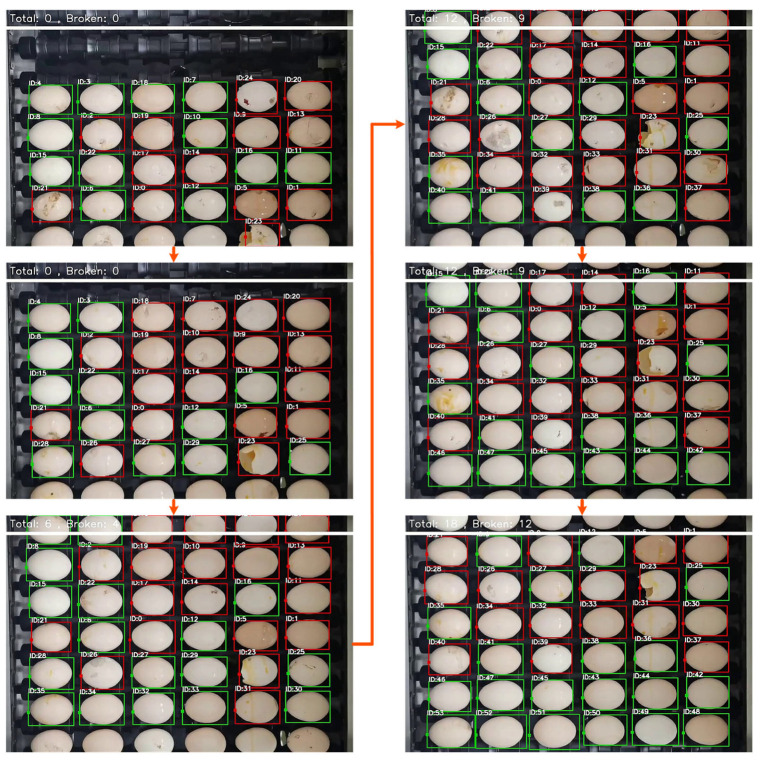
Detection of broken eggs via video.

**Figure 15 foods-12-02179-f015:**
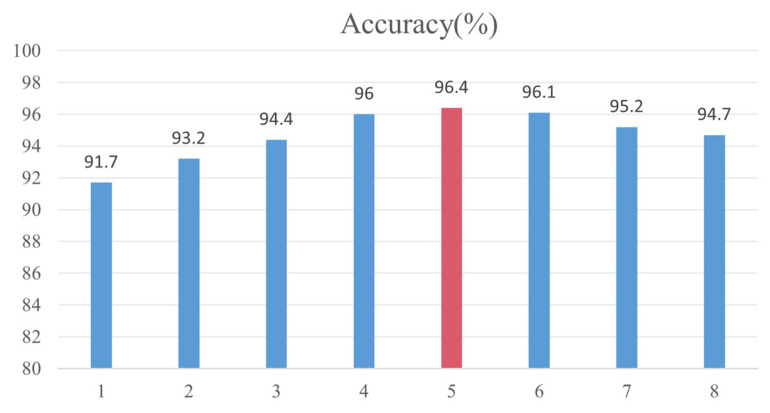
Detection accuracy with different consecutive frame sizes.

**Figure 16 foods-12-02179-f016:**
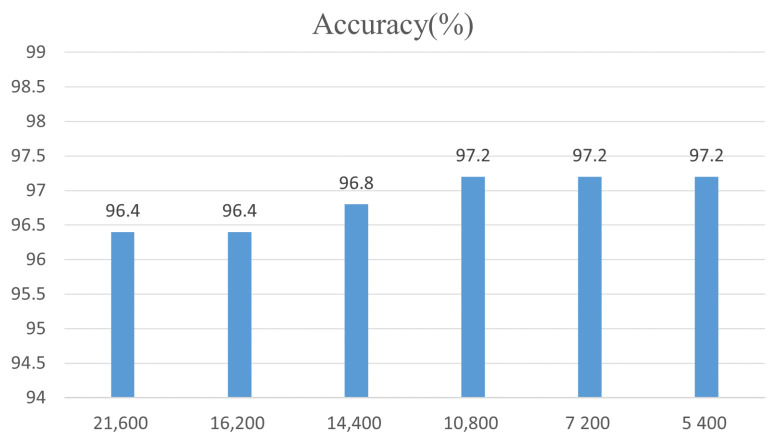
Detection accuracy at different transfer speeds.

**Table 1 foods-12-02179-t001:** Experimental configuration.

Configuration	Parameter
Development environment	Anaconda3-2021.11 + Pycharm
CPU	Intel Core i9-11900 K
GPU	Nvidia GeForce RTX 3060 Ti
Operating system	Windows 10
Accelerated environment	CUDA11.3 CUDNN8.3.0
Development of language	Python3.8

**Table 2 foods-12-02179-t002:** Ablation experiment.

CA	BiFPN	GSConv	Precision (%)	Recall (%)	mAP:0.5 (%)
−	−	−	95.6	90.1	91.1
√	−	−	97.2	92.7	93.4
−	√	−	96.9	92.4	93.3
−	−	√	95.8	91.3	91.8
√	√	√	97.8	94.6	95.2

**Table 3 foods-12-02179-t003:** Comparison of SSD, Faster RCNN, YOLOv4, YOLOv7, and YOLOv5_add.

Model	Class	Precision (%)	Recall (%)	mAP:0.5 (%)	FPS
SSD	Non-broken	89.3	80.4	75.5	22
	Broken	73.1	68		
Faster RCNN	Non-broken	90.8	83.2	79.5	16
	Broken	80.0	77.4		
YOLOv4	Non-broken	97.4	92.2	89.4	31
	Broken	91.2	88		
YOLOv7	Non-broken	99.1	94.8	94.9	37
	Broken	96.8	93.6		
YOLOv5_add	Non-broken	98.9	95.6	95.2	45
	Broken	96.7	93.5		

**Table 4 foods-12-02179-t004:** Actual test performance result.

	Intact (Actual)	Broken (Actual)	Total
Intact (detection)	145	4	149
Broken (detection)	5	98	103
Total	150	102	252

**Table 5 foods-12-02179-t005:** Performance comparison between existing research and the current study.

Researchers	Unwashed Egg	Class			Method	Accuracy
		Intact	Broken	Bloody		
Xiong et al. [15]	−	√	√	−	SVM	97.5%
Tang et al. [18]	−	√	√	−	Improved MobileNetV3	96.3%
Peng et al. [11]	−	√	√	−	WT and BP	90%
Botta et al. [21]	−	√	√	−	SVM and CNN	95.38%
Bao et al. [13]	−	√	√	−	Image Processing	92.5%
Priyadumkol et al. [5]	√	√	√	−	Image Processing	94%
Nasiri et al. [20]	√	√	√	√	Improved VGG16	94.8%
Proposed Model	√	√	√	−	Improved YOLOv5 and ByteTrack	96.25%

## Data Availability

The data presented in this study are available on request from the corresponding author. The data are not publicly available due to partly provided by cooperative enterprises.

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
