# Peer review of "Damage Detection of Unwashed Eggs through Video and Deep Learning"

_foods, 2023, doi:10.3390/foods12112179_

Round 1
Reviewer 1 Report
Manuscript described with the title “Damage detection on unwashed eggs through video and deep learning” has 17 pages, 4 tables and 15 figures. Authors cited 30 references.
Article processing an interesting issue that fully fits into the Scientific Scope of the Foods Journal and Special Issue called “Emerging Processing Technologies: Effects on Improving the Safety and Quality of Food Products”.
Although the manuscript is carefully processed, I still find some recommendations for its improvement. Also, some comments will be important for the article.
Abstract:
Line 11: "Broken eggs are not only harmful to human health, but also unfavorable for transportation and production". I disagree with the claim that "the broken eggs are always harmful" but I recommend that you should change the sentence on: "Broken eggs could be harmful to human health, but also are unfavorable for transportation and production".
Keywords:
Line 27: Add to min. 5 keywords and I recommend to select words that better describe the usability of data from the article for better searching in databases.
Line 35: The names of microorganisms like Salmonella are mentioned in italics.
Introduction and references
A total of 21 references out of 30 total is given in the Introduction chapter. There is a certain disproportion with the other chapters of the text.
Line 216: Figure 5 is not legible.
The text in the chapters of the Results is very technical, yet it is understandable and I have no other comments.
The chapters of Discussion and Conclusions are together, but there are missing discussions in the sense of references.
Some facts are repeated here from the results chapter. I recommend a clear formulation of the benefits for science and the public sector.
Author Response
请参阅附件

Reviewer 2 Report
Dear Authors,
The manuscript (foods-2384645) presented for review is very interesting.
While the manuscript is innovative and interesting, as well as well written, I have a few comments.
Authors, Please note and address the following comments:
Introduction: Background of this study is well written.
Material and methods
Lines 108-121 (in Introduction) are more a methodology than introduction.
Discussion and Conclusion
The weakest part of the manuscript is the Discussion, authors should refer the results of their study with other authors.
The Conclusion should be written as a separate section and authors should indicate direction of future study. Conclusions should answer the questions asked.
What are the authors' recommendations for other scientists?
Limitation
I have a question. Is there any limitation to these results? If yes, it is worth writing about it.
I suggest you to write in the Discussion section about the limitations and strength of these results.
References: References are cited according to journal rules.
Despite my comments, I am pleased to recommend this manuscript for publication. I believe it addresses an important area of research in an international context.
Reviewer
Author Response
请参阅附件。

Reviewer 3 Report
This paper presents an innovative video-based detection model to identify broken eggs in real-time as they move along a conveyor belt in a dynamic environment. The proposed model improves the YOLOv5 object detection algorithm and employs Bytetrack to track eggs in motion, leading to higher detection accuracy compared to traditional single-image based detection models. The authors claim that their improved YOLOv5 model yields a 2.2% increase in precision, 4.4% increase in recall, and a 4.1% increase in mAP:0.5 for detecting broken eggs. The field experiments report an accuracy of 96.4% when using the improved YOLOv5 combined with Bytetrack for video detection of broken eggs.
Although the authors report improvements in various metrics when comparing the improved YOLOv5 model to the original YOLOv5, they do not provide a comparison with other state-of-the-art methods or alternative approaches in egg detection. This omission makes it challenging to gauge the proposed model's performance relative to other potential solutions.
The paper presents two methods for collecting images of unwashed eggs, but it is unclear how many images are collected for each category (intact and broken eggs) and if the dataset is balanced between these two categories. A balanced dataset is crucial for training a reliable and unbiased model.
While the authors mention that the input image resolution for the model is set to 960x960, it is unclear whether the original image resolution of 1920x1080 is resized before feeding it into the model. Resizing images can lead to loss of important details, potentially affecting the model's performance.
The paper does not offer a comprehensive analysis of the effects of different parameters, such as the number of consecutive frames used for egg category determination or variations in conveyor belt speeds. Exploring these factors would provide valuable insights into the model's robustness and applicability under various real-world conditions.
The experiments in this study were conducted on a specific egg processing production line in Guangzhou, Guangdong Province. It remains to be seen if the proposed model would perform equally well in other settings or if it would require further fine-tuning to adapt to different environments.
The paper does not address potential ethical concerns that might arise from the use of such a system, including the potential impact on workers and the broader implications for the egg industry.
The algorithms and the used dataset shall be included as part of the supplementary information and posted publicly in open source data sharing websites. The drawbacks of this study should be included before conclusions section of the draft and it shall outline the variations such as lighting conditions, camera angles and grades of eggs etc and other associates factors.
The language and grammar are ok.
Author Response
请参阅附件。

Round 2
Reviewer 3 Report
the paper can be accepted.